# Springback Control in Complex Sheet-Metal Forming Based on Advanced High-Strength Steel

**Zipeng Lu, Di Li \*, Linlin Cao, Hongjian Cui and Jiachuan Xu**

School of Traffic and Vehicle Engineering, Shandong University of Technology, Zibo 255000, China
* Correspondence: hahali@sdut.edu.cn

**Abstract:** Sheet-metal forming is one of the most important manufacturing processes in the automotive industry. This study proposes a multiobjective optimisation scheme that controls both sheet-metal formability and springback. First, the mechanical properties of DP590 steel were characterised to obtain the mechanical parameters and forming limit diagram (FLD) of DP590. Then, the FLD, thinning rate, and average springback were selected as forming quality evaluation indices. Response surface tests were then conducted for different process parameters for the A-pillar side-stiffener drawing process to analyse the DP590 steel's formability and springback. The optimal process parameters for the drawing process were obtained using a multiobjective optimisation algorithm based on an improved particle swarm method. Finally, a springback compensation scheme was proposed based on the results of the multiprocess simulation. The scheme was applied experimentally to the A-pillar side-stiffener drawing process, and the formability and springback compensation performances verified that the scheme successfully and efficiently eliminated springback and rupture in formed DP590 steel.

**Keywords:** springback; advanced high-strength steel; forming; finite element analysis; multiobjective optimisation



## 1. Introduction

To reduce carbon emissions, lightweighting has become an irreversible trend in the automotive industry—that is, lighter cars can drive further using less energy. Therefore, the pursuit of lightweight components is one of the core objectives of modern transportation engineering [1,2]. Advanced high-strength steel (AHSS), including duplex or dual-phase (DP) steel, has a high strength–ductility balance, so it can be used as automotive steel sheeting. However, there are still many problems associated with the AHSS stamping process, including cracking, wrinkling, scratching, and springback. The first three defects can be eliminated by improving the process conditions, whereas springback is associated with the internal stress-release behaviour of the material itself, which can be difficult to eliminate and can only be controlled. Consequently, stamping to manufacture metal products with exact dimensions and shapes requires the precise control of many variables to control springback [3].

Forming and springback in AHSS are commonly evaluated analytically, experimentally, and by using finite element numerical simulations. In practice, finite element numerical simulations are intuitive and fast and can be used for the study and analysis of stamped parts.

First, accuracy needs to be considered in finite element numerical simulations. Zhang [4] found that the material hardening model had a major influence on the calculation accuracy of sheet-forming springback. Kim [5] combined the advanced intrinsic structure and friction models in the numerical simulation of the bending and forming of TRIP780 steel to ensure greater accuracy of the predicted values of springback and punching forces. Yang [6], Xue [7], and Zajkani [8] all investigated the springback phenomena in the V-bending process

of DP780 steel and the U-bending process of DP500, DP600, and DP780 steel, respectively, considering the unloaded modulus of elasticity. Their results verified that the predicted springback was more accurate when using numerical finite element simulations that considered the unloaded modulus of elasticity.

Second, factors affecting springback need to be considered. Huang [9] analysed the effects of different process parameters on springback during the stamping process using finite element numerical simulations. Nguyen [10] also used finite element simulations to analyse the effects of various factors—such as the blank holder force, friction coefficient, and blank thickness—on springback. Based on the numerical simulation results, it was evident that the blank holder force and blank thickness were the main factors affecting springback. Liu [11] found that a large blank holder force could reduce springback, and that improving the contact between dies could avoid the cracking of stamped parts. Seo [12] found that the radius of the corner of the punch ($R_p$) had a greater effect on springback than the radius of the corner of the die ($R_d$). The springback decreased as $R_p$ increased. Lajarin [13] found the blank holder force to be the most important parameter for springback, followed by the die radius and friction conditions. Chen [14], Andersson [15], and Ozturk [16] investigated the blank holder force, punch fillet radius ($R_p$), and die fillet radius ($R_d$) using numerical analysis, as well as the model gaps, friction coefficients, model shapes, and other factors affecting springback. Starman [17] proposed a numerical method to optimise the blank shape and tool geometry in a 3D sheet-metal-forming operation, the effects of sheet-metal edge geometry and springback after forming and trimming being considered throughout the optimisation process.

Finally, springback compensation measures are essential and can be achieved by carefully designing the shape of the stamping model. Lingbeek [18] proposed two optimisation methods—that is, smooth displacement adjustment (SDA) and surface-controlled overbending (SCO)—both methods use finite element simulations to optimise the tool shape. These approaches have been validated on industrial products. Lee [19] used finite element analysis for parameter optimisation and multilevel compensation to develop an incremental forming process for automotive structural parts using DP980 steel. The simulation results were used to determine the stamping model and stamping-model compensation. After parameter optimisation and multistage model compensation, good dimensional accuracy was obtained.

In this study, a forming and springback optimisation scheme for complex parts was developed. After obtaining the sheet's property parameters and forming limit diagrams (FLDs) through mechanical property tests and establishing the evaluation indices of the forming limit, thinning rate, and springback, the process parameters of the drawing process were considered. The optimal set of process parameters were obtained using an improved particle-swarm-based multiobjective optimisation algorithm to optimise the forming springback. Because only the drawing process was considered during the process-parameter optimisation, springback compensation was applied to the forming tool to reduce the springback so that complex parts with forming limits, thinning rates, and springback could meet the process requirements. To the best of our knowledge, this solution has not yet been tested. This method was applied on a DP590 pressed A-pillar side stiffener, and good springback control performance was verified.

## 2. Experimental Methods

For better simulations of high-strength steel applications with higher prediction accuracy, a material model for the DP590 steel used in actual production was used. The DP590 high-strength steel material model established in this paper is an elastic–plastic model, the mechanical properties of which can be derived through uniaxial tension tests.

Under certain deformation conditions—for example, temperature and deformation rate—the material enters the plastic state only when the individual stress components conform to a certain relationship. This is the yield criterion for the material undergoing plastic deformation. Moreover, when the current stress state point of the material is located

inside the yield surface, the material undergoes recoverable elastic deformation. When the material yields, the size, location, and shape of the yield surface changes with the loading history, the change process being described by the hardening rule. Consequently, the Hill 48 yielding criterion (Equations (1) and (2)) and the Swift hardening criterion (Equation (3)) were chosen for the material model in this study, as follows:

$$F\sigma_{yy}^2 + G\sigma_{xx}^2 + H(\sigma_{xx} - \sigma_{yy})^2 + 2P\sigma_{xy}^2 = 2f(\sigma_{ij}) = \overline{\sigma}^2 \tag{1}$$

$$F = \frac{r_0}{r_{90}(r_0+1)} \quad G = \frac{1}{r_0+1} \quad H = \frac{r_0}{r_0+1}$$
$$L = M = N = \frac{(r_0+r_{90})(1+2r_{45})}{2r_{90}(r_0+1)} \tag{2}$$

where $x$, $y$, and $z$ denote the anisotropy principal axis; $L$, $M$, $N$, $F$, $G$, and $H$ denote the anisotropy parameters; and $r_0$, $r_{45}$, and $r_{90}$ denote the thick anisotropy coefficients of the specimens along the rolling direction of the plate, at an angle 45° to the rolling direction and perpendicular to the rolling direction in three directions. The stress, $\sigma$, is given by:

$$\sigma = K \cdot \left(\varepsilon_p + \varepsilon_s\right)^n \tag{3}$$

where $\varepsilon_p$ denotes the initial strain, $\varepsilon_s$ denotes the plastic strain, $K$ denotes the hardening coefficient, and $n$ denotes the hardening index.

Uniaxial tension tests [20] of the material can be conducted to obtain its mechanical property parameters. The dimensions of the uniaxial tension test and bulging test pieces are shown in Figure 1.

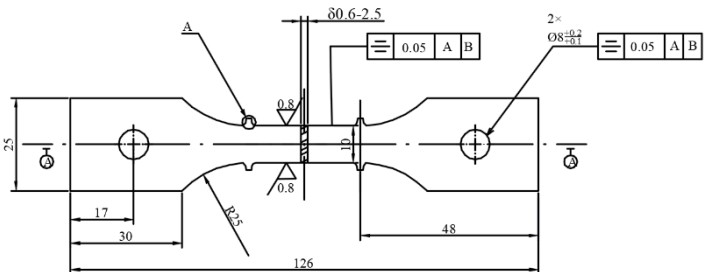

(**a**) Uniaxial tension test specimen and dimensions.

| Dimension | Specimen grade | | | | | | | | |
|---|---|---|---|---|---|---|---|---|---|
| | 1 | 2 | 3 | 4 | 5 | 6 | 7 | 8 | 9 |
| L | 180 | 180 | 180 | 180 | 180 | 180 | 180 | 180 | 180 |
| L1 | 40 | 40 | 40 | 40 | 40 | - | - | - | - |
| L2 | 20 | 20 | 20 | 20 | 20 | - | - | - | - |
| W1 | 40 | 60 | 80 | 100 | 120 | 120 | 120 | 160 | 180 |
| W2 | 30 | 50 | 70 | 90 | 110 | 120 | 140 | 160 | 180 |
| W3 | 20 | 40 | 60 | 80 | 100 | 120 | 140 | 160 | 180 |

(**b**) Bulging test specimens and dimensions.

**Figure 1.** Uniaxial tension test and bulging test pieces (the thickness is 1 mm).

The engineering stress–strain curve of the sheet can be obtained from the results of the uniaxial tension test and the engineering stress–strain can be converted into the real stress–strain using Equation (4) (Figure 2).

$$\begin{cases} \sigma_{nom} = \frac{F}{A} \\ \varepsilon_{nom} = \frac{\Delta L}{L_0} \\ \varepsilon = \ln(1 + \varepsilon_{nom}) \\ \sigma = \sigma_{nom}(1 + \varepsilon_{nom}) \end{cases} \tag{4}$$

where *F* denotes the load, *A* denotes the cross-sectional area of the spar section, $\Delta L$ denotes the length change of the specimen before and after stretching, $\sigma$ and $\sigma_{nom}$ denote the real and engineering stresses, and $\varepsilon$ and $\varepsilon_{nom}$ denote the real and engineering strains.

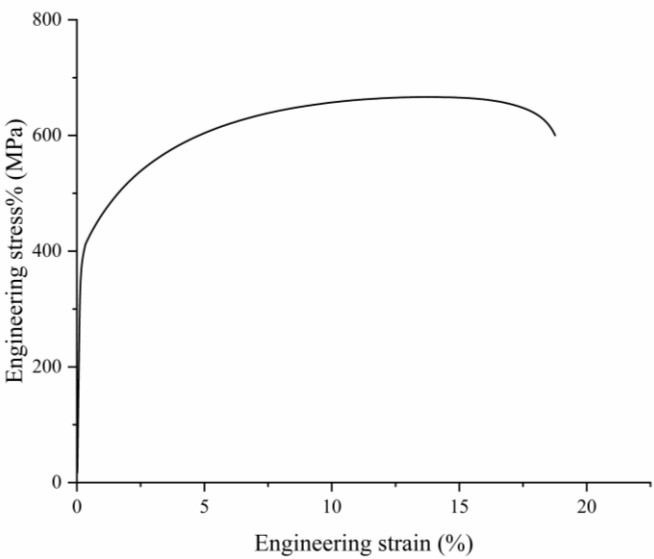

(**a**) DP590 steel engineering stress–strain curve

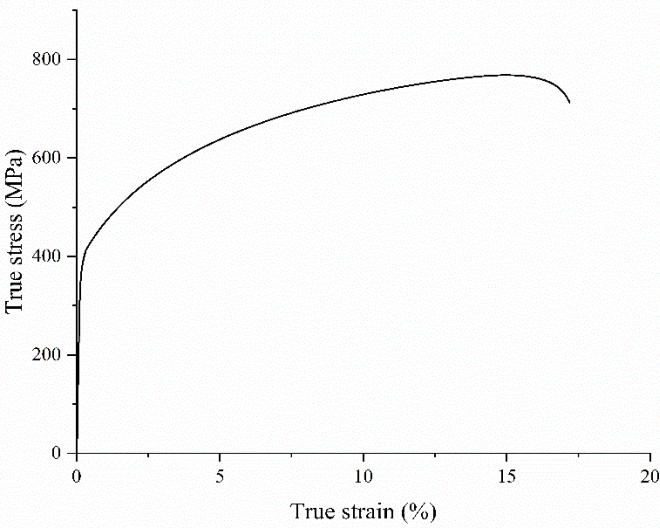

(**b**) DP590 steel real stress–strain curve

**Figure 2.** DP590 steel engineering stress–strain curve and real stress–strain curve.

The material parameters can be obtained through parameter fitting, using the Swift model (Equation (3)), the values of which can be calculated by combining Equations (1) and (2), as shown in Table 1.

**Table 1.** Material parameters of the DP590 specimens.

| Young's Modulus $E$/MPa | Poisson Ratio $u$ | Hardening Coefficient $K$ | Hardening Index $n$ | Anisotropic Parameters | | | Elongation (%) | Thinning Rate $T$ (%) [21] |
|---|---|---|---|---|---|---|---|---|
| | | | | $r_0$ | $r_{45}$ | $r_{90}$ | | |
| 201,000 | 0.28 | 950 | 0.179 | 0.71 | 0.96 | 0.71 | 22 | 30 |

The FLD can be determined using the following procedure [22]:

Step 1: Establish the strain coordinate system by taking the surface strain $\varepsilon_2$ abscissa and surface strain $\varepsilon_1$ as ordinates.

Step 2: Plot the surface limit strain values ($\varepsilon_2$, $\varepsilon_1$), which are measured through experiments in the strain coordinate system.

Step 3: According to the distribution characteristics of the surface limit strains in the coordinate system, connect these points to the appropriate curves. The curve is called the forming limit curve and the coordinate system is called the forming limit diagram.

The results of the expansion test are integrated into the FLD by first fitting the $FLD_0$ points of the sheet and then forming the FLD from the data points of the expansion test. Finally, the FLD can be expressed as follows:

$$\begin{cases} FLD_0 = 1.656n + 0.032 - 0.025 \\ \varepsilon_1 = FLD_0 + \varepsilon_2(nd_1\varepsilon_2 + d_2) \end{cases} \tag{5}$$

where $n$ denotes the steel hardening index, and $d_1$ and $d_2$ denote the parameters to be fitted.

The DP590 steel FLD obtained is shown in Figure 3 and the FLD parameters of the DP590 steel specimens are listed in Table 2.

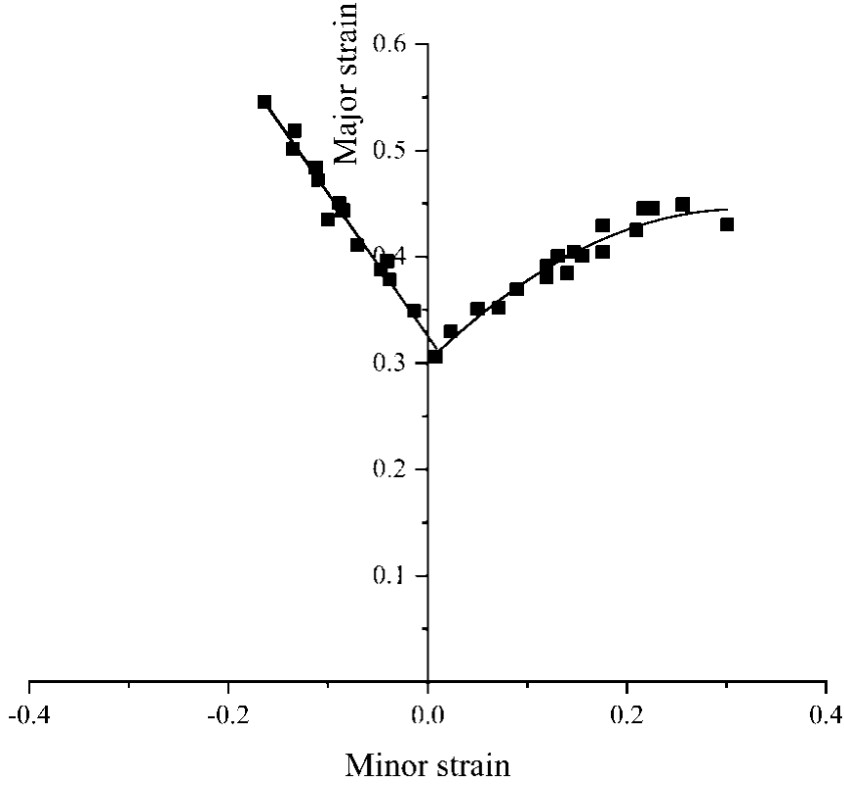

**Figure 3.** The DP590 steel forming limit diagram (FLD).

**Table 2.** The FLD parameters of the DP590 steel specimens.

| $n$ | $d_1$ | | $d_2$ | | $R^2$ |
|---|---|---|---|---|---|
| | Left | Right | Left | Right | |
| 0.179 | 4.01184 | −6.9519 | −1.3138 | 0.7920 | 0.960 |

### 3. Results: Forming Quality Evaluation

The FLD can be used in finite element simulations to evaluate the forming quality of parts, although there is a disadvantage in that the FLDs can only view the forming situation through cloud diagrams and cannot compare the effects of different process parameter combinations on the forming quality, nor can they determine the optimal union process parameter sets. Because this study considered the evaluation of spring-back, the evaluation criteria of high-strength-steel forming quality, considering both forming quality and springback accuracy, were proposed based on the FLD, thinning rate, and the amount of springback, so that different forming and springback results could be quantitatively compared.

#### 3.1. The Forming Limit Diagram (FLD)

Combined with the FLD of DP590 steel that was obtained from the expansion test, the safety margin determines the degree of cracking risk of the material. The critical cracking line and risk of cracking can be generated in the FLD based on the safety margin (Figure 4). The safety margin is 30%. The area between the two is the risk-of-crack zone. Consequently, no grid is allowed to exist in the crack zone. In addition, as few grids as possible are in the risk-of-crack zone in the forming quality evaluation.

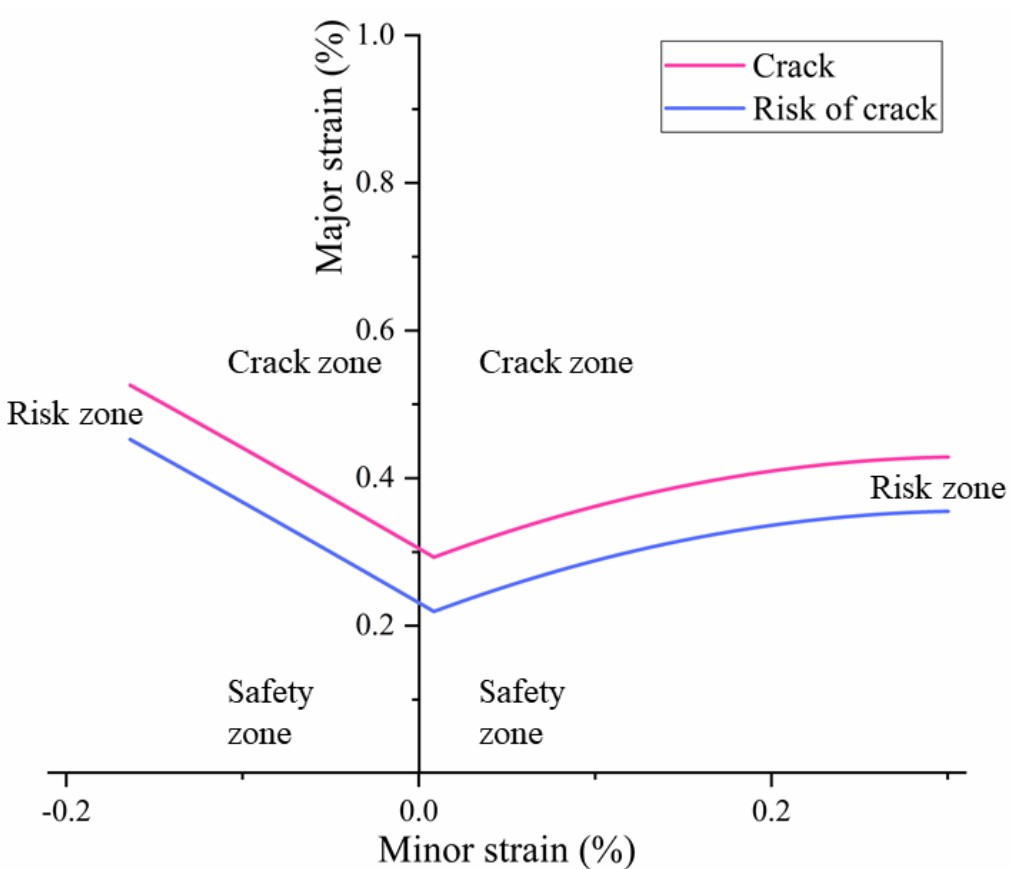

**Figure 4.** The cracking and risk-of-cracking lines for DP590 steel.

After importing the material parameters into the DYNAFORM simulation software, the forming limit cloud diagram of the forming results can be viewed in the postprocessing software, the forming quality being evaluated using the different colour representations in the diagram—that is, the FLD evaluation index—as shown in Figure 5.

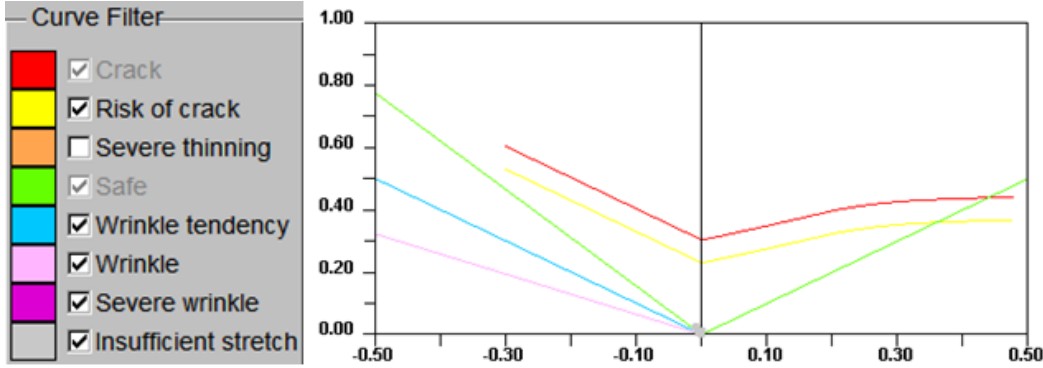

**Figure 5.** The FLD of DP590 steel in DYNAFORM.

### 3.2. Evaluation of the Thinning Rate

The thinning rate ($T$) is the ratio of the sheet thickness before and after forming. From the thinning-rate cloud diagram, whether there is more thinning or more thickening can be determined, with too much thinning or thickening resulting in poor forming. The thinning rate measured in the previous chapter represents the limit thinning rate of a tensile specimen when it breaks. When evaluating the stamping of complex parts, the maximum thinning rate is 20%. The maximum thickening rate [23,24] is usually 20%. In this paper, because of complex sheets, the maximum thickening rate is 10%.

### 3.3. Springback Evaluation

After unloading the sheet-forming load, the elastic recovery of the sheet results in its shape and size changing in the opposite direction from the loading deformation. This phenomenon is called springback. Springback can be positive or negative and will affect the forming accuracy. Consequently, the absolute value of the sum of the "average springback" evaluation, can be expressed as follows:

$$S_a = \frac{\left(S_z + \left|S_f\right|\right)}{2} \tag{6}$$

where $S_z$ denotes the maximum value of positive springback, $S_f$ denotes the maximum value of negative springback, and $S_a$ denotes the average of the maximum positive and maximum negative springback values.

Because complex parts need accurate dimensions, in this study, the springback evaluation range was within ±0.5 mm, according to the process requirements.

Finally, the forming result needs to simultaneously meet the three requirements of forming limit, thinning rate, and the amount of springback. First, there is no cracking in the FLD. Then, the thinning rate and amount of springback need to meet the production requirements.

## 4. Finite Element Numerical Simulation Design

The A-pillar side stiffener of a commercial vehicle is a complex stamped part (Figure 6), which is located between the inner and outer plates of the A-pillar stiffener. Combining the features and production scheme, the process flow can be defined in four stages—that is, the drawing; trimming and punching; flanging and shaping; and punching and side-punching processes. The drawing process is the first and most important step and has a great influence on the forming quality and springback.

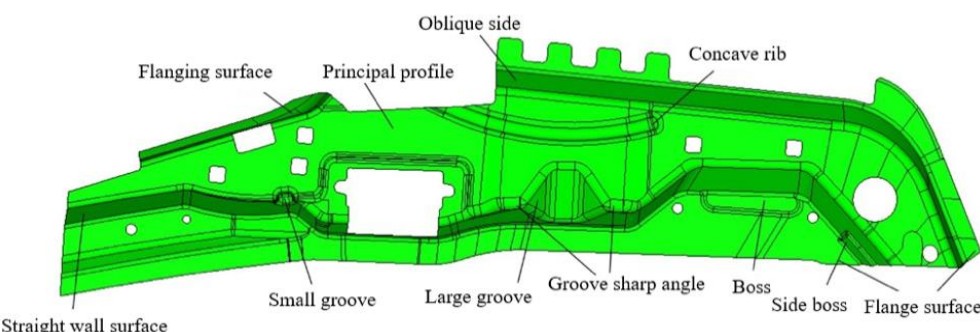

**Figure 6.** Drawing of the A-pillar side stiffener.

DYNAFORM is a software that provides integrated CAE solutions for stamping products and dies. DYNAFORM was chosen for this study because it can predict cracks, wrinkles, thinning, scratches, and springback forming stiffness, and provide surface evaluations of sheet metal formability.

The overall size of the A-pillar side stiffener is about 920 mm × 222 mm × 0.9 mm. In the DYNAFORM finite element simulation, the A-pillar stiffener inner plate is divided into shell elements with a size of 8 mm and a total of 10,233 mesh elements. The mesh selection element type of sheet metal division in the drawing process is the shell element, and the mesh element size is 8 mm, with a total of 5838 mesh elements. The sheet metal is consistent with the material in Section 2. The thickness of DP590 steel is 1 mm.

Therefore, this study established a numerical simulation model for the drawing process (Figure 7) and analysed the influence of each process parameter on the forming quality and springback in the drawing process itself.

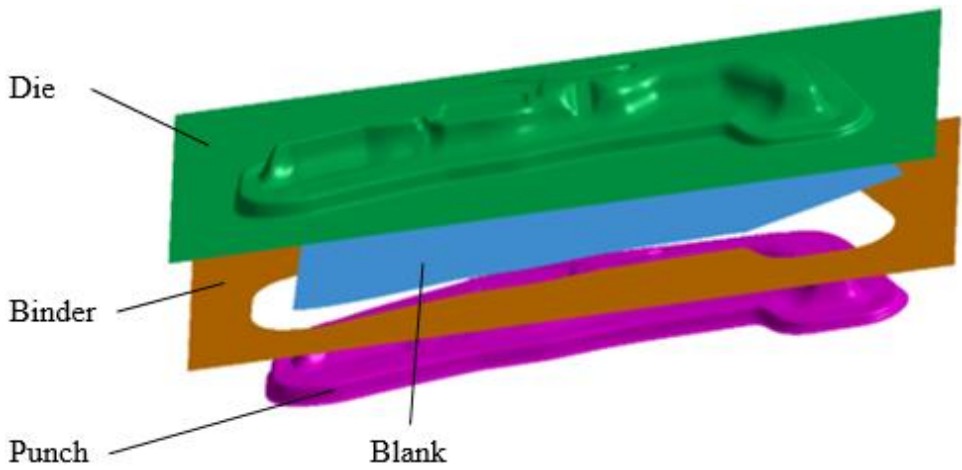

**Figure 7.** The forming model of the drawing process.

*4.1. Influencing Factors*

(1)   Friction coefficient

The friction coefficient is usually large to reduce the amount of springback, but at the same time, it can be very easy to damage the plate material's surface or facilitate cracking; if the friction coefficient is smaller, the opposite occurs. The friction coefficient, $\mu$, is expressed in Equation (7):

$$\mu = \frac{F}{F_N} \tag{7}$$

where $F$ denotes the friction pressure, and $F_N$ denotes the pressure.

(2)   Model clearance

A small clearance increases the friction between the sheet and the model, causing faster model wear and increased manufacturing costs, potentially affecting the quality of the sheet forming, increasing the forming force of the model on the sheet, and making the sheet rupture due to it being pulled beyond its material limits during the forming process. When the clearance is larger, the friction between the sheet and the model is reduced, and the risk of rupture is reduced; however, the plastic forming area may also be reduced, especially when advanced high-strength steel materials are used for stamping and forming, the underforming and springback phenomena becoming more serious. Consequently, in the initial simulation, the model clearance of the sheet was determined to be 1.1 times the material thickness.

(3)   Blank holder force

By applying axial pressure to the contact area between the blank holder surface and the concave model, the flow direction of the sheet can be controlled to ensure smooth sheet forming. If the blank holder force is too large, the sheet could easily break; if the blank holder force is too small, wrinkling defects may be produced as the model fails to compress the sheet and becomes unstable. In this study, the value of $P$ was set to be 3 MPa. The blank holder force, $F$, in N, is expressed as:

$$F = AP \tag{8}$$

where $A$ denotes the blank projection area, and $P$ denotes the blank holder force per unit area (MPa).

(4)   Punch fillet radius

When stamping, the force on the sheet near the corner of the model is concentrated. When the punch fillet radius of the model changes, the range and magnitude of the force on the sheet also changes, affecting the forming defects and springback. In this study, the radii of the corner of the model were set to 5, 7, and 9 mm.

*4.2. Response Surface Test Design*

Considering that both the thinning rate and average springback are quantitative evaluation indices, the response surface test is often used to study the influence of multiple factors on the stamping results during finite element simulations. Consequently, the orthogonal test was designed to study the effects of different levels of process parameters on the A-pillar side stiffener in the drawing process, as summarised in Table 3. In Table 3, $C_o$ is the factor level, $F_{blk}$ is the blank holder force (KN), $R_p$ is the punch fillet radius (mm), $f$ is the friction coefficient, and $X$ is the model clearance (mm). A total of twenty-seven sets of response surface tests(Table 4) were then designed, with four factors and three levels.

**Table 3.** Factor level table.

| $C_o$ | $F_{blk}$ | $R_p$ | $f$ | $X$ |
|---|---|---|---|---|
| −1 | 1200 | 5 | 0.10 | 0.9 |
| 0 | 1300 | 7 | 0.12 | 0.99 |
| 1 | 1400 | 9 | 0.14 | 1.08 |

Based on the above results, a quadratic polynomial with cross terms could be established to fit the prediction model. The least-squares method was used to solve for the

unknown to obtain the response models for the drawing thinning rate (*T*) and the average springback (*S_a*), as follows:

$$
\begin{aligned}
T = {} & 1.155 - 5.2 \times 10^{-4} F_{blk} - 0.002.3 \times 10^{-3} R_p - 8.028 f - 0.490 X \\
& + 6 \times 10^{-8} F_{blk}{}^2 + 3.63 \times 10^{-4} R_p{}^2 + 20.51 f^2 + 7.27 \times 10^{-2} D^2 \\
& - 9 \times 10^{-7} F_{blk} R_p + 2.317 \times 10^{-3} F_{blk} f + 1.681 \times 10^{-4} F_{blk} X \\
& - 3.6 \times 10^{-3} R_p f - 1.06 \times 10^{-3} R_p X + 1.11 f X
\end{aligned}
\tag{9}
$$

$$
\begin{aligned}
S_a = {} & -25.3 + 2.87 \times 10^{-2} F_{blk} + 0.736 R_p - 9.3 f + 17.9 X \\
& + 9 \times 10^{-6} F_{blk}{}^2 - 1 \times 10^{-4} R_p{}^2 + 233 f^2 + 14.14 X^2 \\
& - 2.98 \times 10^{-4} F_{blk} R_p - 3.03 \times 10^{-2} F_{blk} f - 2.742 \times 10^{-2} F_{blk} X - 1.52 R_p f \\
& - 0.062 \times 10^{-2} R_p X - 34.7 f X
\end{aligned}
\tag{10}
$$

**Table 4.** The response surface test scheme and results.

| Number | Variable Coded Value | | | | Experiment Result | |
|---|---|---|---|---|---|---|
| | $F_{blk}$ | $R_p$ | $f$ | $X$ | $T$ | $S_a$ |
| 1 | −1 | −1 | 0 | 0 | 18.111 | 3.8725 |
| 2 | 1 | −1 | 0 | 0 | 19.556 | 3.6485 |
| 3 | −1 | 1 | 0 | 0 | 18.088 | 4.5055 |
| 4 | 1 | 1 | 0 | 0 | 19.464 | 4.043 |
| 5 | 0 | 0 | −1 | −1 | 17.873 | 4.4385 |
| 6 | 0 | 0 | 1 | −1 | 21.356 | 3.038 |
| 7 | 0 | 0 | −1 | 1 | 17.208 | 5.52 |
| 8 | 0 | 0 | 1 | 1 | 21.490 | 3.8695 |
| 9 | −1 | 0 | 0 | −1 | 18.195 | 3.472 |
| 10 | 1 | 0 | 0 | −1 | 19.223 | 3.4405 |
| 11 | −1 | 0 | 0 | 1 | 17.903 | 5.2515 |
| 12 | 1 | 0 | 0 | 1 | 19.536 | 4.233 |
| 13 | 0 | −1 | −1 | 0 | 17.390 | 4.6235 |
| 14 | 0 | 1 | −1 | 0 | 17.666 | 5.0785 |
| 15 | 0 | −1 | 1 | 0 | 21.490 | 3.1805 |
| 16 | 0 | 1 | 1 | 0 | 21.708 | 3.392 |
| 17 | −1 | 0 | −1 | 0 | 17.345 | 4.786 |
| 18 | 1 | 0 | −1 | 0 | 17.686 | 4.6775 |
| 19 | −1 | 0 | 1 | 0 | 20.218 | 3.4645 |
| 20 | 1 | 0 | 1 | −1 | 22.413 | 3.1135 |
| 21 | 0 | −1 | 0 | −1 | 18.643 | 3.397 |
| 22 | 0 | 1 | 0 | 1 | 18.697 | 3.8175 |
| 23 | 0 | −1 | 0 | 1 | 18.825 | 4.2495 |
| 24 | 0 | 1 | 0 | 0 | 18.803 | 4.6255 |
| 25 | 0 | 0 | 0 | 0 | 18.575 | 3.9795 |
| 26 | 0 | 0 | 0 | 0 | 18.577 | 3.9642 |
| 27 | 0 | 0 | 0 | 0 | 18.587 | 3.949 |

$F_{blk}$: blank holder force (KN); $R_p$: punch fillet radius (mm); $f$: friction coefficient; $X$: model clearance (mm); $T$: thinning rate (%); $S_a$: average springback.

From Figure 8, it is evident that the actual values of the drawing thinning rate and the average springback have a linear relationship with the predicted values. It can be concluded that the predicted values of the fitted model are more accurate, and the mathematical model can be used to replace the stamping model for the next analysis.

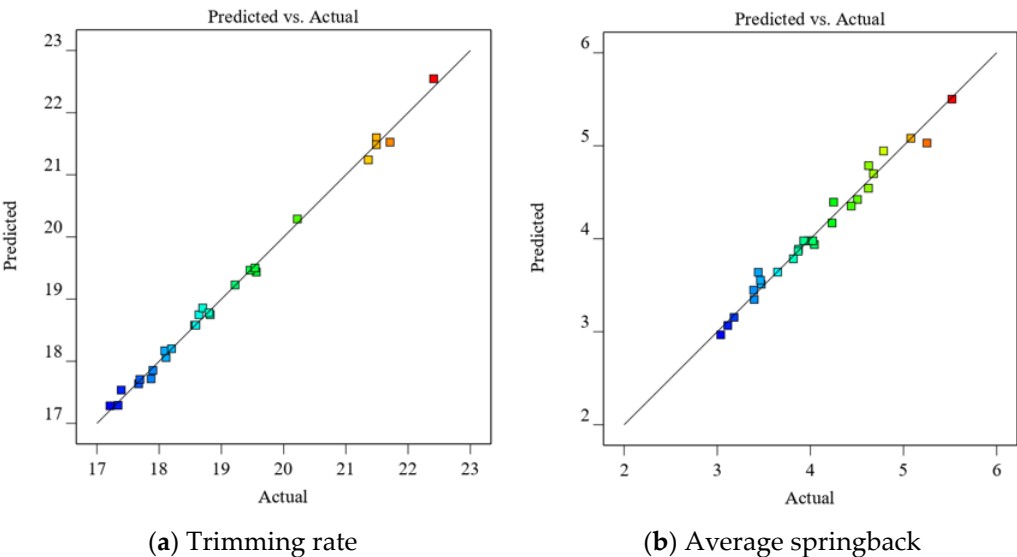

(**a**) Trimming rate           (**b**) Average springback

**Figure 8.** Comparison between the actual values and predicted values.

## 5. Discussion: Stamping Result Optimisation of the Drawing Process

The optimisation process can be divided into two parts. That is, the process parameters of the drawing process can be optimised to obtain the optimal set of process parameters, after which the springback compensation can be conducted based on the springback results to ensure that the final stamping results meet the requirements of the three evaluation indices (forming limit, thinning rate, and amount of springback).

### 5.1. Multiobjective Optimisation Based on Improved Particle Swarm Algorithm

The particle swarm optimisation (PSO) algorithm is an intelligent optimisation algorithm based on the predatory behaviour of birds. Compared to other algorithms, it requires fewer parameters to be modified, is easy to implement, and is computationally efficient, and the best particles can be searched based on individual and global information [25], which plays an important role in the application of multiobjective optimisation. The optimisation process itself is as shown in Figure 9.

PSO treats each individual as a particle and uses position velocity and fitness values to represent the relevant characteristics of the particles by selecting individual extremum and global extremum, again and again, to update the velocity and position of particles. The particle update equation can be expressed as follows:

$$V_{id}(k+1) = V_{id}(k) + c_1 r_1 (P_{id}(k) - X_{id}(k)) + c_2 r_2 (P_{gd}(k) - X_{gd}(k))$$
$$X_{id}(k+1) = X_{id}(k) + V_{id}(k+1) \tag{11}$$

where $c_1$ and $c_2$ denote acceleration coefficients, $r_1$ and $r_2$ denote random numbers between [0, 1], $V_i$ and $X_i$ denote the velocity and position of the $i$th particle, and $P_i$ and $P_g$ denote the individual and global extremes.

The inertia weight ($\omega$) can be invoked in the velocity update formula, and $\omega$ determines whether the particle is performing a velocity update or accepting the current velocity. In this paper, $\omega$ can be expressed as follows:

$$\omega(i) = \omega_{end} \times (\omega_{start} / \omega_{end})^{1/(1+i/g_{\max})} \tag{12}$$

where $i$ denotes the current number of iterations, $\omega_{start}$ denotes the initial inertia weight, $\omega_{end}$ denotes the inertia weight when iterating to the maximum number, and $g_{\max}$ denotes the maximum number of iterations.

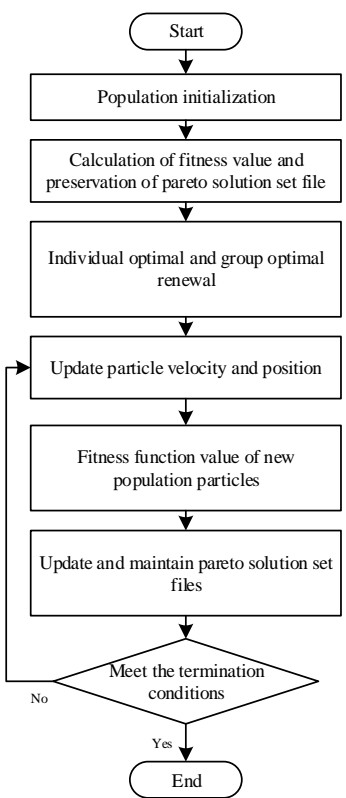

**Figure 9.** Flow chart of the multiobjective optimisation process.

The springback and thinning ratio after forming should be as small as possible when optimising the drawing process parameters of the A-pillar side stiffener. Therefore, the two response surface models should be minimized. The established multiobjective optimisation model of the drawing process for the A-pillar stiffener is shown in Equation (13).

$$odj(\min T(\%), \min S_a(mm))$$

$$s.t. \begin{cases} 1200 \le F_{blk} \le 1400 \\ 5 \le R_p \le 9 \\ 0.11 \le f \le 0.15 \\ 0.9 \le X \le 1.08 \end{cases} \tag{13}$$

The Pareto optimisation solution set shown in Figure 10 can be obtained by using the improved particle swarm algorithm for the response surface model with particles as design variables—that is, $X = [F_{blk}, R_p, f, X]$, a maximum number of 100 iterations, and an initial population of 100.

Because the forming precision is the most important problem in the forming of complex parts, in the Pareto optimised solution sets, the minimum springback is given priority when selecting the optimal process parameter group. As such, the minimum draw thinning rate is 18.812% and the average springback is 3.154 mm. The optimal combination of process parameters is a 0.9 mm model clearance, a 5 mm model fillet radius, a 0.124 friction coefficient, and a 1240.76 KN blank holder force. Evidently, the thinning rate meets requirements, but the springback is still too large.

The finite element numerical simulation of the drawing process showed that the thinning rate (Figure 11) met the requirements with no ruptures or wrinkling phenomena (Figure 12), indicating good optimisation of the thinning rate. Consequently, the next step needs to be optimised for springback compensation.

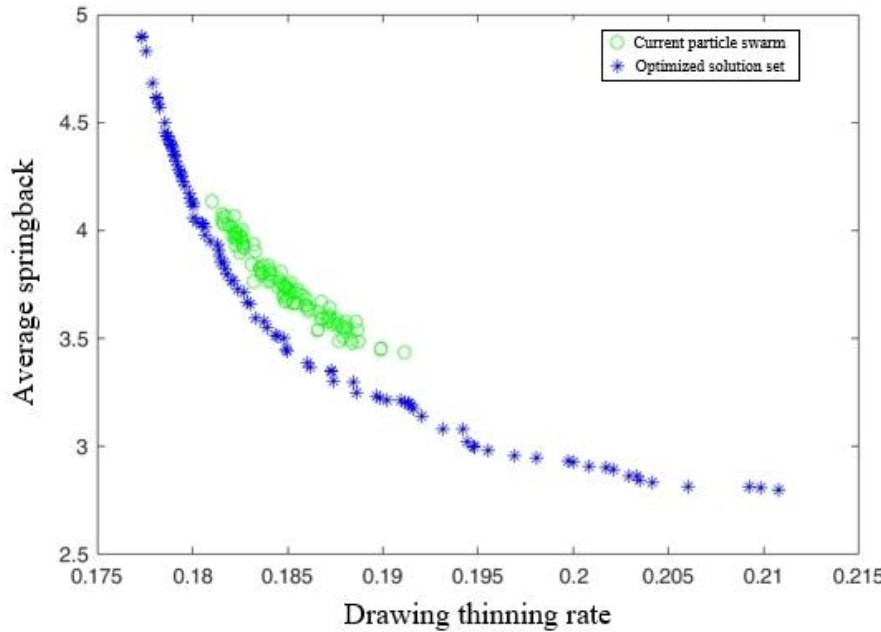

**Figure 10.** Pareto optimised solution sets.

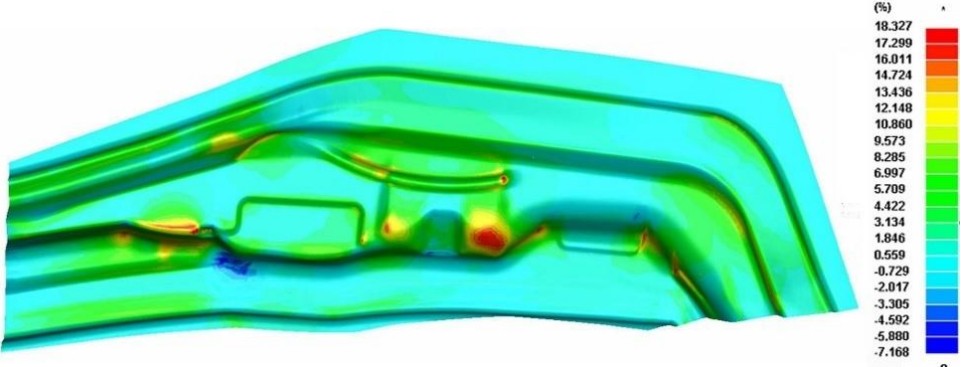

**Figure 11.** Thinning rate diagram of the optimised drawing process.

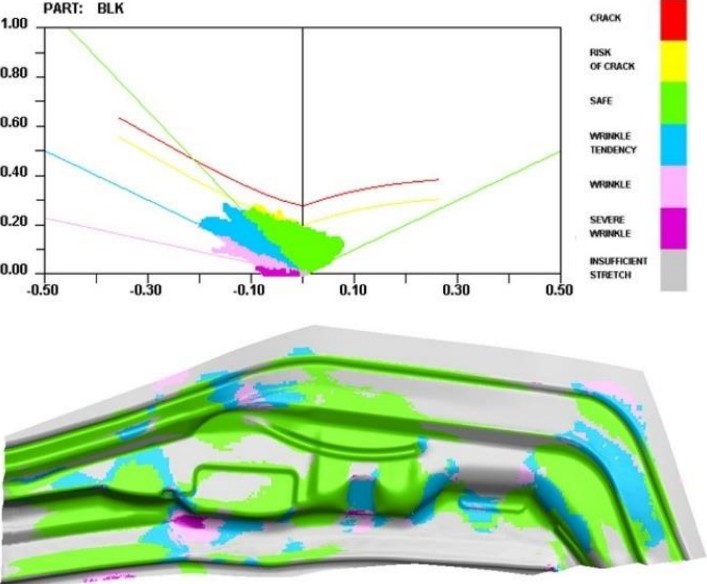

**Figure 12.** Drawing and forming quality diagram of the optimised drawing process.

### 5.2. Springback Compensation

In the previous section, the four processes for the A-pillar side stiffener were defined as being drawing; trimming and punching; flanging and shaping; and punching and side-punching. The previous section only optimised the process parameters based on the drawing process; however, the flap shaping can also produce elastic deformation, and the trimming and punching and the punching and side-punching processes can also produce springback due to the strong frictional force of the tool on the edge of the part. Consequently, the springback compensation scheme proposed in this study considered all processes.

The springback compensation process proposed this study compensated the drawing model surface by using the difference between the trimmed edge and the original model. It also compensated the flanged forming model surface by using the difference between the springback result of the side punching and the nonspringback result after the trimming process. The optimisation process is shown in Figure 13.

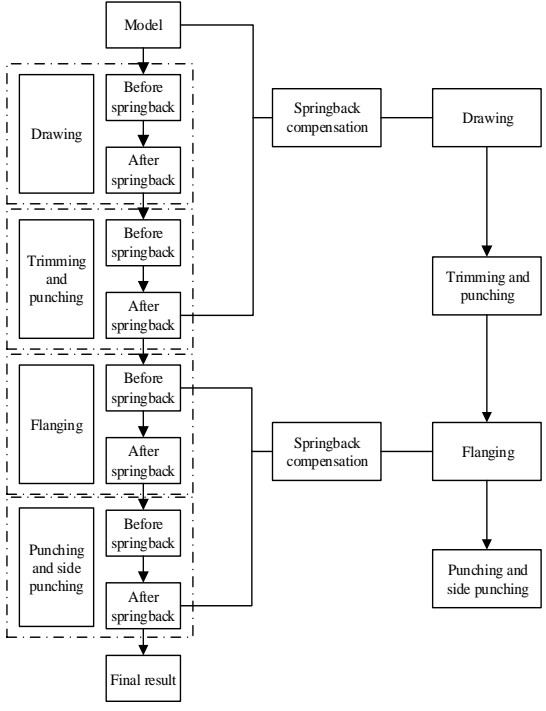

**Figure 13.** Flow chart of the springback compensation process.

The compensation steps, in detail, are as follows:

Step 1: Numerical finite element simulation of the entire process is conducted in DYNAFORM, with corresponding numerical simulations of springback being conducted after each process.

Step 2: Springback compensation for the drawing process, the shape before springback being the original model and the shape after springback being the shape after springback of the trimming and punching process.

Step 3: Numerical simulation of the forming finite elements and springback of the entire process, again.

Step 4: Springback compensation for the flanging and shaping process, the shape before springback being the shape before the springback of the flap-shaping process, and the shape after springback being the result after the springback of the punching and side-punching process.

From Figure 14, it is evident that the final formed A-pillar side stiffener does not exhibit cracking, and there is no serious wrinkling on the surface that fits with the inner

and outer A-pillar. From Figure 15, it is evident that the maximum thinning rate is 19.838%, which does not exceed the required thinning rate of 20%. From Figure 16, it is evident that the maximum positive springback of the final product is 0.394 mm, and the maximum negative springback is −0.492 mm, meeting the requirements.

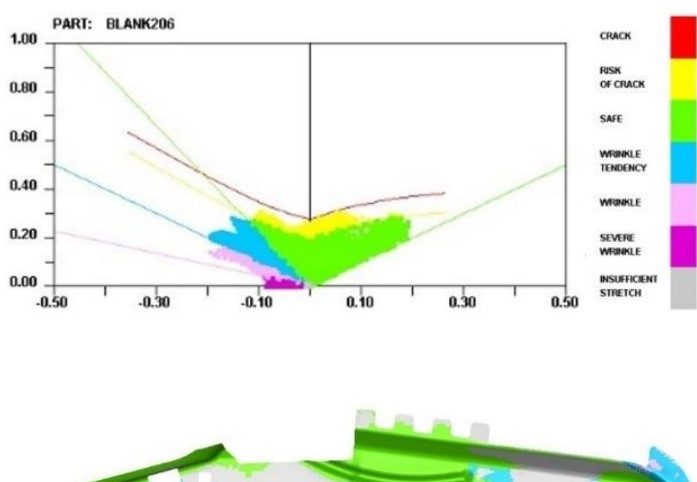

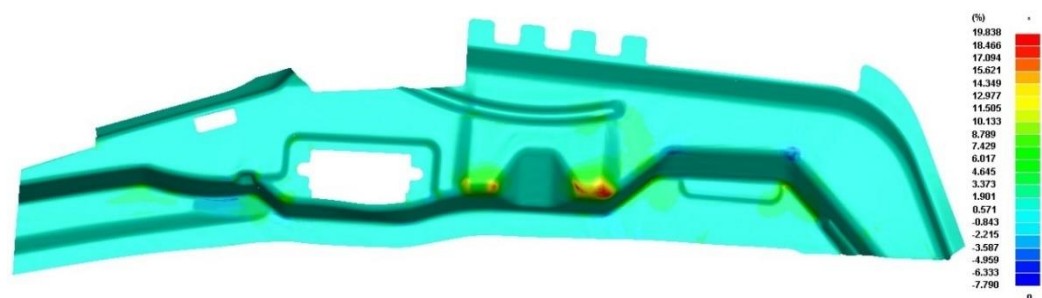

**Figure 14.** Forming limit of the final product.

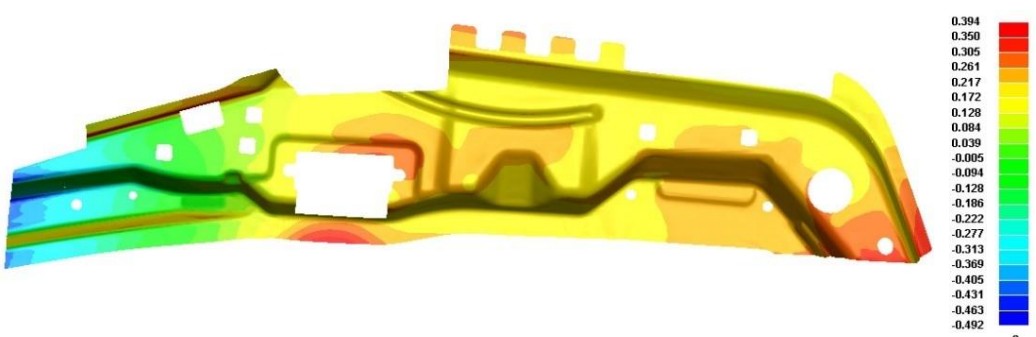

**Figure 15.** Thinning rate of the flanged shaped product.

**Figure 16.** Resilience of the final product.

In summary, the A-pillar side stiffener, after optimisation of the process parameters, was subjected to multiprocess numerical simulations to obtain the amount of springback for each process, and based on the simulation results, a springback compensation strategy was determined. Then, based on the determined springback compensation scheme, a springback compensation simulation was conducted and verified in DYNAFORM. The

results indicated that the final product after springback compensation was within the specified 20% thinning limits. The maximum positive springback value was 0.394 mm, and the maximum negative springback value was −0.492 mm, which satisfied the springback limit requirement of ±0.5 mm. Moreover, the forming quality was good. Consequently, the results showed the springback compensation strategy developed in this study to be effective based on the results of the multiprocess springback simulation after optimisation of the process parameters.

*5.3. Experimental Results Verification*

The final, pressed, formed part, when the optimal parameter values were used in the compensated model, is as shown in Figure 17. After checking, the overall dimensional accuracy requirements were measured using professional measuring instruments. When measuring the thickness of the area with the most serious thinning, the minimum thickness is 0.83 mm, and the minimum thinning rate is required. It is evident that the thickness was uniform, and there was no apparent breakage or wrinkling. Therefore, good formability and minimal springback were achieved, and the high forming accuracy verified the correctness of the simulation analysis, as well as the optimisation calculations.

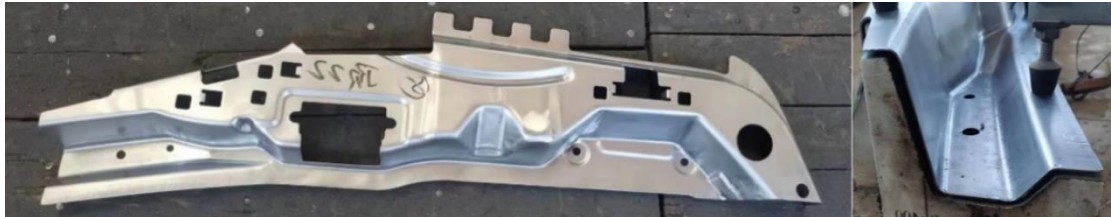

**Figure 17.** Actual formed part.

**6. Conclusions**

When stamping complex parts using advanced high-strength steel, forming and springback problems occur. In this study a forming optimisation scheme for complex parts that considers the forming limit, thinning rate, and average springback was developed. The following key points are drawn from the study:

(1) Uniaxial tension tests at room temperature were conducted on DP590 steel sheeting to obtain the mechanical property parameters and FLD of the material and to establish the evaluation indices of the stamping and forming quality. The FLD can be used to evaluate the forming quality of the part to eliminate ruptures. Consequently, the thinning rate and the average amount of springback met the corresponding production requirements.

(2) A response surface model was established, and the mathematical relationships between the process parameters of the drawing process, the thinning rate, and the average springback were established. The optimal process parameter set was determined using the improved particle-swarm-based multiobjective optimisation algorithm, and the process parameter set with the minimum springback value that guarantees drawing formability was obtained.

(3) Based on the process requirements, a multiprocess springback compensation model that fully considers the springback of each process was developed to ensure that the results met the process requirements. After test stamping, it was evident that the formability was good, no rupture and wrinkling occurred, and the amount of springback was small. These results verified the accuracies of the numerical simulation and optimisation scheme.

**Author Contributions:** Conceptualization, D.L. and Z.L.; methodology, Z.L.; software, L.C.; validation, Z.L.; formal analysis, L.C. and Z.L.; investigation, H.C.; resources, D.L.; data curation, L.C.; writing—original draft preparation, Z.L.; writing—review and editing, D.L.; supervision, D.L. and J.X. All authors have read and agreed to the published version of the manuscript.

**Funding:** This project was supported by the China-Sino Truck Group's industry–university–research cooperation program (project No. 20200810).

**Institutional Review Board Statement:** Not applicable.

**Informed Consent Statement:** Not applicable.

**Data Availability Statement:** Not applicable.

**Conflicts of Interest:** The authors declare no conflict of interest.

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
