# Peer review of "Springback Control in Complex Sheet-Metal Forming Based on Advanced High-Strength Steel"

_coatings, doi:10.3390/coatings13050930_

Round 1

Reviewer 1 Report

The manuscript is focused on the forming and springback issues of stamping process of complex parts using advanced high strength steel. A springback compensation model was proposed and successfully validated for a stamping process of an A-pillar stiffener part made from DP590 steel sheet.

The paper is interesting and fits the aim and scope of the journal.

Major problems:

1.      English must be revised.

2.      The equations must be correctly numbered.

3.      Usually, in the Swift hardening model  denotes the initial strain and  denotes the plastic strain.

4.      Which standard was followed in the uniaxial tension tests? And in the bulging tests?

5.      Figure 2.2 only shows the real stress-strain curve but in the manuscript you refer that Figure 2.2 shows the engineering stress-strain curve and real strain diagram (line 126).

6.      Do you refer to equation 2.3 in line 126 and 135?

7.      Can you explain this sentence: “The uniaxial tension test is performed by shearing the parts at different angles”.

8.      Can you explain the procedure to determine the forming limit diagram? The used methodology must be presented and detailed described in the manuscript.

9.      Why did you select the DynaForm simulation software?

10.  Which was the criteria to select 10% thickening and 20% thinning as limits in the thinning rate section?

11.  The section finite element numerical simulation design is the number 4.

12.  More details about the numerical model must be added to the manuscript (element type, number of elements…).

13.  In Figure 5.6 there are points of deformation near the crack limit and the risk of crack limit disappearing. Can you explain why?

14.  In line 365 you refer that the springback limit is satisfied. Where did you define the limit of 0.5 mm?

15.  In section 5.4 more results must be presented. Thickness measurements must be presented and compared with numerical predicted thickness. Crucial dimensions (lengths and angles) should be compared with numerical results.

16.  The thickness of the DP590 steel sheet must be identified.

Author Response

Response to Reviewer 1 Comments

Response to Reviewer 1 Comments

Point 1: English must be revised.

Response 1: English has been revised, such as Abstract and Introduction.

Point 2: The equations must be correctly numbered.

Response 2: We have chescked and removed the number of equations.

Point 3:Usually, in the Swift hardening model denotes the initial strain and denotes the plastic strain.

Response : The editor told us, the missed parts are not important, they will remove them.

Point 4:Which standard was followed in the uniaxial tension tests? And in the bulging tests?

Response : We have changed some details. The uniaxial tension tests refer to GB/T 228.1-2010 Metallic materials-Tensil testing-Part 1: Method of test at room temperature; The bulging tests refer to GB/T 15825.8-2008 Sheet metal formability and test methods-Part 8: Guidelines for the determination of forming-limit diagrams.(Line 117,143).

Point 5:Figure 2.2 only shows the real stress-strain curve but in the manuscript you refer that Figure 2.2 shows the engineering stress-strain curve and real strain diagram (line 126).

Response : We have provided the engineering stress-strain curve in this article.(Line 132-137).

Point 6:Do you refer to equation 2.3 in line 126 and 135?

Response : In the new article, we have changed this low-level error,that is in line 139.

Point 7:Can you explain this sentence: “The uniaxial tension test is performed by shearing the parts at different angles”.

Response : This sentence is used in the draft of the paper, The real meaning of the right scentence is “The material parameters can be obtained by parameter-fitting using the Swift model. The values of which can be calculated by combining equations in the paper” (Line138-140)

Point 8:Can you explain the procedure to determine the forming limit diagram? The used methodology must be presented and detailed described in the manuscript.

Response : Based on GB/T 15825.8-2008, the procedure is shown in the new paper.(Line 143)

Point 9:Why did you select the DynaForm simulation software?

Response : DYNAFORM is a software that provides CAE integrated solution technology for stamping products and dies. The reason why DYNAFORM is chosen in this paper is that Dynaform can predict cracks, wrinkles, thinning, scratches, springback forming stiffness, surface evaluation of sheet metal formability, so as to provide help for sheet metal forming process and die design. (Line223-226)

Point 10:  Which was the criteria to select 10% thickening and 20% thinning as limits in the thinning rate section?

Response : We read lots of literature, the thickening and thinning rate is usually the optimization objective, the thinning rate is usually below 30% and the thickening rate is usually below 20%, because of the complex sheet forming, we choose the thinning rate below 20% and the thickening rate below 10% to avoid defect.(Line 191-197)

Point 11:The section finite element numerical simulation design is the number 4.

Response : There was an oversight in the framing of our article. It has been changed to “4. Finite Element Numerical Simulation Design(Line 214)

Point 12:More details about the numerical model must be added to the manuscript (element type, number of elements…).

Response : We provide the elements type , it is Shell element,the element is the quadrilateral mesh with side length of 8mm,and the number of elements are also provided in the paper.(Line 227-232)

Point 13: In Figure 5.6 there are points of deformation near the crack limit and the risk of crack limit disappearing. Can you explain why?

Response : The Expression is imprecise in 3.1 , The grid in the risk zone should be as less as possible instead of no grid is allowed to exist in the crack-risk zone or crack zone in the forming quality evaluation. What’s more, based on GB/T 15825.8-2008 and engineering experience, the safety margin of FLD in this paper is safer than usual grids.(Line 179-180)

Point 14:In line 365 you refer that the springback limit is satisfied. Where did you define the limit of 0.5 mm?

Response : This is a high precision parts,the limit is based on the accuracy requirement and the requirements of enterprises.(Line 208-209)

Point 15: In section 5.4 more results must be presented. Thickness measurements must be presented and compared with numerical predicted thickness. Crucial dimensions (lengths and angles) should be compared with numerical results.

Response : We ignored the process, so we contact the enterprises to get some details. We measure the size of the sheet and some thickness, and the measurements all meet the requirements

Point 16:The thickness of the DP590 steel sheet must be identified.

Response : All of the thickness of the DP590 steel sheet is 1mm, this detail is changed.(Line 124, 232)

Reviewer 2 Report

The main question addressed by the research is about a new springback optimisation scheme for complex part, the optimal set of process parameters being obtained using an improved particle-swarm-based multi-objective optimisation algorithm to optimise the forming springback.

I am considering that the topic of springback analysis is common in the field but the optimization scheme based on particle-swarm-based multi-objective algorithm is new.

The authors made a complete analysis of the springback, starting with the material characterization, then they obtained the FLD curve for the material. Using a simulation program the authors evaluated the material forming quality. Finite Element Numerical Simulation is used to study the deep drawing formability of the A-pillar side stiffener of a commercial vehicle process. For this, they used in the first step the response surface method. They considered the variation of four parameters at three levels each: blank holder force; punch fillet radius; friction; clearance. As outer parameters were considered the thinning and springback. They obtain a mathematical model which the authors used in the next step of their study which was the multi-objective optimisation based on improved particle swarm algorithm. They propose a scheme for the springback compensation process, which include the model obtained by optimization, and was used in A-pillar side stiffener quality improvement, by FEM. Finally they verifying the correctness of the simulation analysis as well as the optimisation calculations with an experiment work.

                 The methodology about applying the particle swarm algorithm must be more detailed.  Which are the parameters A, B, C, D from equation 5.3 (or 0.13). What is the connection between the particle swarm algorithm and Pareto optimization?

The conclusions are clear.

              The references are appropriate.

There are some problems with the equation numbers which are not correct. Also, the numbers of figures must be in order, from the beginning of the paper till the end.

Finally, I didn’t understand why the paper was sent to Coating journal, in condition in which the subject has no connection with the journal topics.

The paper must be revised.

Author Response

Response to Reviewer 2 Comments

Point 1: The methodology about applying the particle swarm algorithm must be more detailed.Which are the parameters A, B, C, D from equation 5.3 (or 0.13). What is the connection between the particle swarm algorithm and Pareto optimization? 

Response 1: In order to reflect optimization more clearly, we changed A,B,C,D to Fblk, Rp, f, X. (Line 331).

It can be seen that there are lots of parameters can fit the scope of thinning rate and average springback value together,but cant get the minimum value,so Pareto optimization is to get the best status. Apparently, because of forming accuracy, average springback value is more important,that is the strategy to get the best parameters. In conclusion, the particle swarm algorithm is the strategy to get the scope, and Pareto optimization is used to analyze the best parameters.(Line 338-344)

Point 2:  I didn’t understand why the paper was sent to Coating journal, in condition in which the subject has no connection with the journal topics.

Response 2: When we looked up the literature, we found this journal and tried to contact the editor. The editor expressed interest in this paper and encouraged us to contribute

Round 2

Reviewer 1 Report

The paper is interesting and fits the aim and scope of the journal.